# Multi-Plane Program Induction with 3D Box Priors

**Yikai Li**[1,2*]    **Jiayuan Mao**[1*]    **Xiuming Zhang**[1]
**William T. Freeman**[1,3]    **Joshua B. Tenenbaum**[1]    **Noah Snavely**[3]    **Jiajun Wu**[4]

[1]MIT CSAIL        [2]Shanghai Jiao Tong University
[3]Google Research        [4]Stanford University

## Abstract

We consider two important aspects in understanding and editing images: modeling regular, program-like texture or patterns in 2D planes, and 3D posing of these planes in the scene. Unlike prior work on image-based program synthesis, which assumes the image contains a single visible 2D plane, we present Box Program Induction (BPI), which infers a program-like scene representation that simultaneously models repeated structure on multiple 2D planes, the 3D position and orientation of the planes, and camera parameters, all from a single image. Our model assumes a *box prior*, i.e., that the image captures either an *inner view* or an *outer view* of a box in 3D. It uses neural networks to infer visual cues such as vanishing points or wireframe lines to guide a search-based algorithm to find the program that best explains the image. Such a holistic, structured scene representation enables 3D-aware interactive image editing operations such as inpainting missing pixels, changing camera parameters, and extrapolate the image contents.

## 1   Introduction

We aim to build autonomous algorithms that can infer two important structures for compositional scene understanding and editing from a single image: the regular, program-like texture or patterns in 2D planes and the 3D posing of these planes in the scene. As a motivating example, when observing a single image of a corridor like the one in Fig. 1, we humans can effortlessly infer the camera pose, partition the image into five planes—including left and right walls, floor, ceiling, and a far plane—and recognize the repeated pattern on each of these planes. Such a holistic and structural representation allows us to flexibly edit the image, for instance by inpainting missing regions, moving the camera, and extrapolating the corridor to make it infinite.

A range of computer vision algorithms have utilized such a holistic scene representation to guide image manipulation tasks. Several recent ones fit into a program-guided image manipulation framework [11, 24, 25]. These methods infer a program-like image representation that captures camera parameters and scene structures, enabling image editing operations guided by such programs so that the scene structure is preserved during editing. However, due to the combinatorial complexity of possible compositions of elementary components based on the program grammar, these methods usually only work for images in highly specific domains with a fixed set of primitives such as hand-drawn figures of simple 2D geometric shapes [11] and synthesized tabletop scenes [24], or natural images with of a *single* visible plane, such as ground tiles and patterned cloth [25, 21].

To address these issues and scale up program-guided image manipulation, we present a new framework, namely, Box Program Induction (BPI, for short), that jointly segments the image into multiple planes and infers the repeated structure on each plane. Our model assumes a *box prior*, leveraging the observation that box-like structures widely exist in images. Many indoor and outdoor scenes fall into this category: walking in a corridor or room corresponds to observing a box from the inside, and taking a picture of a building corresponds to seeing a box from the outside.

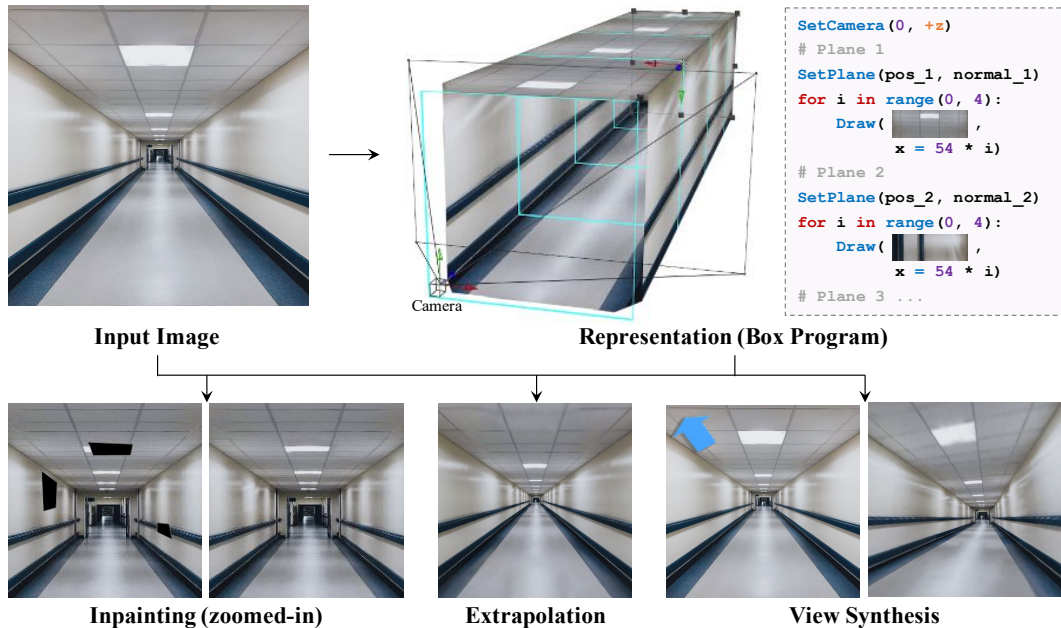

Figure 1: We present Box Program Induction (BPI), which infers a program-like scene representation that simultaneously models repeated structure on *multiple* 2D planes, 3D positions and orientations of the planes, relative to the camera, all from a *single* image. The inferred program can be used to guide perspective- and regularity-aware image manipulation tasks, including image inpainting, extrapolation, and view synthesis.

To enable efficient inference of box programs, we also propose to utilize mid-level cues, such as vanishing points or wireframe lines, as well as high-level visual cues such as subject segmentations, to implicitly constrain the search space of candidate programs. Given the input image, BPI first infers these visual cues with pre-trained data-driven models. Next, it enumerates all candidate programs that satisfy the implicit constraints imposed by these inferred visual features. Finally, it ranks all candidate programs using low-level visual features such as pixel reconstruction.

In summary, we present BPI, a framework for inducing *box programs* from images by exploiting learned visual cues. Our experiments show that BPI can efficiently and accurately infer the structure and camera parameters for both indoor and outdoor scenes. The inference procedure is robust to errors and noise inherent to visual cue prediction: BPI automatically selects the best candidate wireframe lines and refines the vanishing points if they are not accurate. BPI also enables users to make 3D-aware interactive editing to images, such as inpainting missing pixels, extrapolating image content in specific directions, and changing camera parameters.

## 2 Related Work

**Visual program induction.** Computer graphics researchers have used procedural modeling (mostly top-down) for representing 3D shapes [19, 33] and indoor scenes [35, 20, 29]. With advances in deep networks, some methods have paired top-down procedural modeling with bottom-up recognition networks. Such hybrid models have been applied to hand-drawn sketches [11], scenes with simple 2D or 3D geometric primitives [31, 24], and markup code [9, 5]. The high-dimensional nature of procedural programs poses significant challenges to the search process; hence, even guided by neural networks, these works focus only on synthetic images in constrained domains. SPIRAL [12] and its follow-up SPIRAL++ [26] use reinforcement learning to discover latent "doodles" that are later used to compose the image. Their models work on in-the-wild images, but cannot be directly employed in tasks involving explicit reasoning, such as image manipulation and analogy making, due to the lack of program-like, interpretable representations.

In the past year, Young et al. [41] and Mao et al. [25] integrated formal programs into deep generative networks to represent natural images, and later applied the hybrid representation to image editing. Li et al. [21] extended these models by jointly inferring perspective effects. All these models, however, assume a single plane in an image, despite the fact that most images contain multiple planes such as floor and ceiling. Our BPI moves beyond the single-plane assumption by leveraging box priors.

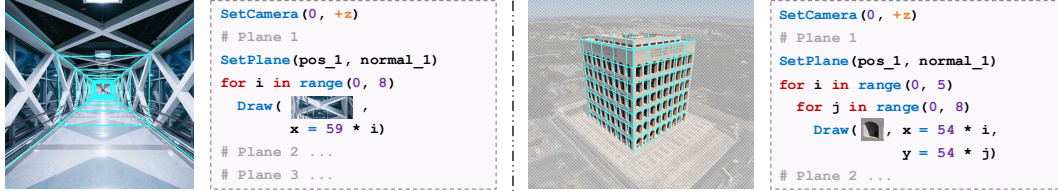

<div align="center">

(a) Box Program for Corridors        (b) Box Program for Buildings

</div>

Figure 2: Example box programs inferred by BPI. Our box program jointly models camera parameters, the 3D positions and orientations of multiple planes, as well as the regularity structure on individual planes.

| |
|---|
| Program⟶CameraProgram; WorldProgram; |
| CameraProgram⟶SetCamera(pos=Vec3, point_to=Vec3); |
| WorldProgram⟶PlaneProgram; \| PlaneProgram; World Program; |
| PlaneProgram⟶SetPlane(pos=Vec3, normal=Vec3) For1Stmt; |
| For1Stmt⟶For ( $i$ in range(Integer, Integer) ){ DrawStmt; \| For2Stmt } |
| For2Stmt⟶For ( $j$ in range(Integer, Integer) ){ DrawStmt } |
| DrawStmt⟶Draw ($x$=Expr, $y$=Expr) |
| Expr⟶Real $\times i$ + Real $\times j$ \| Real $\times i$ |

Table 1: The domain-specific language (DSL) of box programs. Language tokens `For`, `If`, `Integer`, `Real`, and arithmetic/logical operators follow the Python convention. `Vec3` denotes 3D real vectors.

**Image manipulation.** Image manipulation, in particular image inpainting, is a long-standing problem in graphics and vision. Pixel-based [1, 3] and patch-based methods [10, 4, 13] achieve impressive results for inpainting regions that requires only local, textural information. They fail on cases requiring high-level, structural, or semantic information beyond textures. Image Melding [8] extends patch-based methods by allowing additional geometric and photometric transformations on patches, but ignores global consistency among patches. Huang et al. [16] also use perspective correction to assist patch-based inpainting, but rely on vanishing points detected by other methods. In contrast, BPI segments planes and estimates their normals based on the global regularity of images.

Advances in deep networks have led to impressive inpainting algorithms that integrate information beyond local pixels or patches [17, 40, 42, 23, 43, 45, 39, 38, 28]. In particular, deep models that learn from a single image (a.k.a., internal learning) produce high-quality results for image manipulation tasks such as inpainting, resizing, and expansion [32, 45, 30]. BPI also operates on a single image, simultaneously preserving the 3D structure and regularity during image manipulation.

## 3 Box Program Induction

Our proposed framework, Box Program Induction (BPI), takes an image as input and infers a box program that best describes the image, guided by visual cues.

### 3.1 Domain-Specific Language

We start by describing the domain-specific language (DSL) that we use to express the multiple planes in the scene and the regularity structure on individual planes, namely the *box programs*. We assume these planes are faces of a 3D box. Take Fig. 2a as an example: the corridor is composed of four planes, each containing visually regular patterns.

Table 1 specifies the DSL, and Fig. 2 shows sample programs. A box program consists of two parts: camera parameters and programs for individual planes. A *plane program* first sets the plane's surface normal, then specifies a sub-program defining the regular 2D pattern on the plane. These patterns utilize the primitive `Draw` command, which places patterns at specified 2D positions. `Draw` commands appear in `For`-loop statements that characterize the regularity of each plane.

### 3.2 Box Program Fitness

Given an input image, our goal is to segment the image into different planes, estimate their surface normals relative to the camera, and infer the regular patterns. We treat this problem as seeking a program $P$ that best fits the input image $I$. We first define the *fitness* of a program $P$ by measuring how well $P$ reconstructs $I$. Recall that a box program $P$ is composed of multiple *plane programs*, each of which defines the regular pattern on a plane as well as its position and orientation in 3D. Combined with camera parameters, we can use parameters of each plane program to *rectify* the corresponding plane, resulting in images without perspective effects $\{J_1, J_2, \cdots, J_k\}$, where $k$ is the number of planes described by $P$.

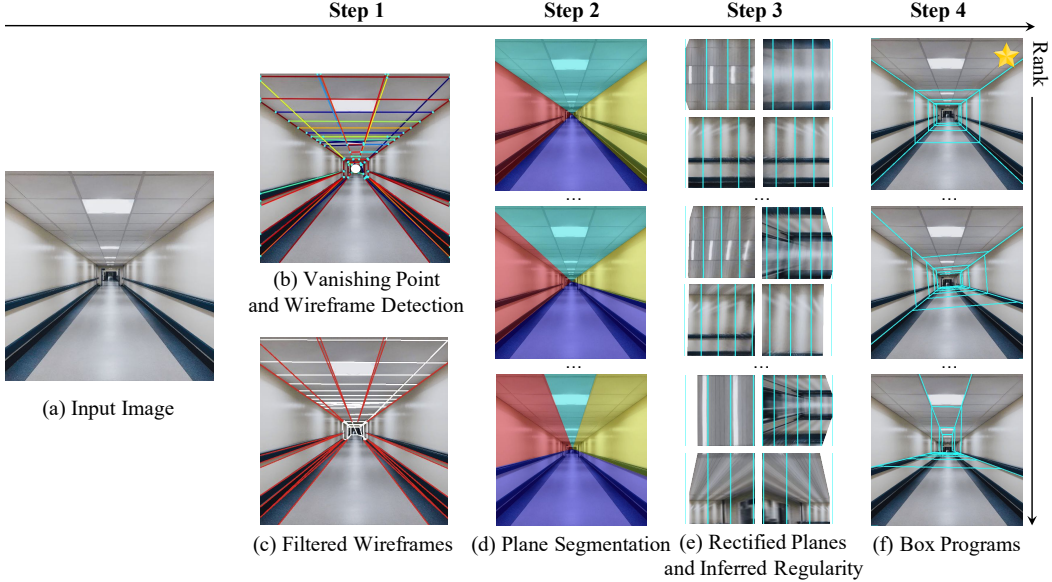

Figure 3: Our Box Program Induction finds the best-fit program that describes the input image (a). It first detects the vanishing point and wireframe segments (b), followed by a filtering step (c). It then constructs a set of candidate plane segmentation maps (d). Given each plane segmentation, it rectifies each plane and infers its regularity structure (e). We rank all candidate box programs by their fitness (f); the starred candidate is the best.

For each plane $i$, we compute its fitness score by comparing its rectified image $J_i$ and the corresponding plane program block, denoted as $Q_i$. The fitness score is defined in a similar way as in the Perspective Plane Program Induction framework [21]. Specifically, executing $Q_i$ produces a set of 2D coordinates $C_i$ that can be interpreted as the centers of each visual element, such as the center of the lights on the ceiling in Fig. 2a. Since $Q_i$ contains a nested loop of up to two levels, we denote the loop variable for each For loop as $a$ and $b$. Thus, each 2D coordinate in $C_i$ can be written as a function of $a$ and $b$, $c(a, b) \in \mathbb{R}^2$. The fitness score $F$ is defined as

$$F = -\sum_{a,b} \left[ \|J_i[c(a,b)] - J_i[c(a+1,b)]\|_2^2 + \|J_i[c(a,b)] - J_i[c(a,b+1)]\|_2^2 \right], \quad (1)$$

where $\| \cdot \|_2$ is the $\mathcal{L}_2$ norm and $J_i[c(a,b)]$ denotes the patch centered at 2D coordinates $c(a, b)$. Since we only consider lattice patterns on individual planes, we implement this by first shifting $J_i$ leftward (or downward) by the 'width' (or 'height') of a lattice cell and then computing the pixel-wise difference between the shifted image and the original image $J_i$. The overall program fitness of $P$ is the sum of the fitness function for all planes.

### 3.3 3D Box Priors and the Role of Visual Cues

A naïve way to find the best-fit program $P$ is to enumerate all possible plane segmentations of the input image $I$ and rank all candidates by the fitness score (Eq. 1). However, the complexity of this naïve method scales exponentially in the number of planes. Instead, we propose to consider first, the *box prior*, which constrains the plane segmentation of the image and, second, visual cues that help to guide the search. Specifically, we impose the following *box prior*:

- For an *inner view* of a box, e.g., a corridor as in Fig. 2a, our box program models four planes: two side walls, the floor, and the ceiling. For images with a far plane, we will segment the far plane but do not use programs to model it, as most far planes do not contain a regular structure.
- For an *outer view* of a box, e.g., a building as in Fig. 2b, our box program models the two side walls as two planes. We do not model the roof as, in most images, the roof is either nonvisible or very small in size.

Below, we show how to use *visual cues* to guide the search for box programs. We focus on the *inner view* case, and include details for the *outer view* case in the supplemental material. The full search process is illustrated in Fig. 3, and consists of four steps. First, we use pre-trained neural networks to detect the vanishing point and the 2D wireframes line segments from the image. We also filter out invalid wireframe segments. Next, we generate a set of candidate plane segmentations based on

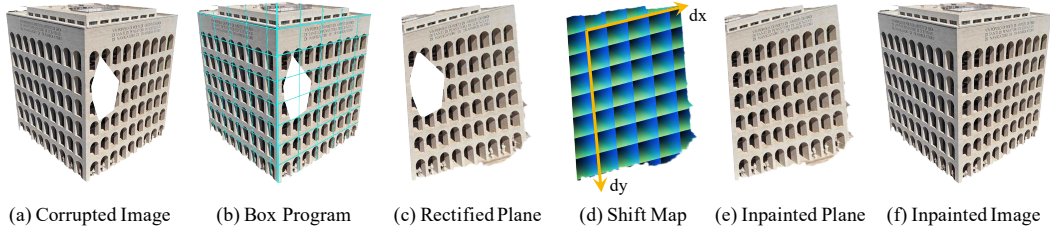

(a) Corrupted Image    (b) Box Program    (c) Rectified Plane    (d) Shift Map    (e) Inpainted Plane    (f) Inpainted Image

Figure 4: An illustration of the proposed program-guided PatchMatch. Given the corrupted image, we first detect its box program (b) and rectify each plane (c). We use the regular structure (d) on the plane to guide the PatchMatch to inpaint the corrupted plane (e). The color in (d) visualizes the relative position of each pixel to the center of its associated lattice cell.

the detected wireframe segments. Then, for each segmented plane, we seek the program that best describes the plane structure. Finally, we rank all candidate plane segmentations by the fitness score.

**Step 1: Visual cue detection.** Following the *box prior*, all inner views of a box contain a single *vanishing point* and four *intersection lines* that are the intersection between the two walls, the ceiling, and the floor in 3D. These intersection lines will be projected onto the image plane as four lines that intersects at the vanishing point. We use this property to constrain the candidate plane segmentation. Leveraging vanishing point information for inferring box structures of scenes has also been studied in [15].

Given an input image (Fig. 3a), we apply NeurVPS [46] to detect the vanishing point and L-CNN [47] to extract wireframes in the image. We use the most confident prediction of NeurVPS as the vanishing point $vp \in \mathbb{R}^2$, which is a 2D coordinate on the image plane. Each wireframe segment is represented as a segment $AB$ on the image, from $(x^A, y^A)$ to $(x^B, y^B)$, as illustrated in Fig. 3b. Next, we filter out wireframe segments whose length is smaller than a threshold $\delta_1$ or whose extension does not cross a neighbourhood centered at $vp$ with radius $\delta_2$. The remaining wireframe segments, denoted by the set $WF$, are illustrated in Fig. 3c.

**Step 2: Plane segmentation.** We then enumerate all combinations of four wireframe segments from $WF$. As these wireframe segments $s_i = [(x_i^A, y_i^A), (x_i^B, y_i^B)], i = 1, 2, 3, 4$ may not intersect at a single point, we compute a new vanishing point $vp^*$ that minimizes $\sum_i dist(vp^*, s_i)$, where $dist$ is the distance between the point $vp^*$ and the line containing segment $s_i$. Next, we connect the new vanishing point $vp^*$ and the farther end of each $s_i$ to get four rays. These four rays partition the image into four parts, which we treat as the segmentation of four planes, as illustrated in Fig. 3d.

**Step 3: Plane rectification and regularity inference.** Fixing the camera at the world origin, pointing in the $+z$ direction, we then compute the 3D position and surface normal of each plane. As shown in Fig. 1, because the distance between camera and the corridor is coupled with the focal length of the camera, here we use a fixed focal length of $f = 35mm^*$. Based on these assumptions, the four rays can *unambiguously* determine the 3D positions and surface normals of four planes. The proof can be found in the supplemental material.

Based on the inferred surface normal, we can *rectify* each plane, yielding a set of rectified images $\{J_1, J_2, \cdots, J_4\}$. For each rectified plane, we search for the best plane program that describes it, based on the fitness function Eq. 1. The inferred plane regularity structures are shown in Fig. 3e.

**Step 4: Box program ranking.** We sum up the fitness score for four planes in each candidate segmentation as the overall program fitness. We use this score to rank all candidate segmentations, and choose the program with the highest fitness to describe the entire image.

## 3.4 Program-Guided Image Manipulation

The inferred box program enables 3D-aware interactive image manipulation: inpainting missing pixels, changing camera parameters , and extrapolating the image content. Lying at the core of these applications is a program-guided PatchMatch algorithm.

Our program-guided PatchMatch augments the PatchMatch algorithm [4] by enforcing the regularity on each plane. Consider the task of inpainting missing pixels on the wall of a building in Fig. 4.

Taking a corrupted image as input (Fig. 4a), we infer the box program from the undamaged pixels (Fig. 4b). The inferred box program explains the plane segmentation and the regularity pattern on each plane. For each plane that contains missing pixels, we can first rectify the corrupted plane as shown in Fig. 4c. Since the plane program on this plane will partition the image with a lattice, we construct a "shift map", illustrated as Fig. 4d, which represents the relative position of each pixel to the center of the lattice cell that this pixel lies in, normalized to $[-0.5, 0.5]$.

In PatchMatch, the similarity $sim(p, q)$ between pixel $p$ and pixel $q$ is computed as a pixel-level similarity $sim_{\text{pixel}}$ between two patches centered at $p$ and $q$, with a constant patch size $\delta_{\text{pm}}$. We add another term to this similarity function:

$$sim(p, q) \triangleq sim_{\text{pixel}} + sim_{\text{reg}} = sim_{\text{pixel}} - \lambda_{\text{reg}} \|\text{wraparound}(smap[p] - smap[q])\|_2^2, \quad (2)$$

where $\lambda_{\text{reg}}$ is a hyperparameter that controls the weight of the regularity enforcement. abs is the absolute value function. The shift-map term measures whether two pixels $p$ and $q$ correspond to the same location on two (possibly different) repeating elements on the plane. We "wrap around" shift map distance by $\text{wraparound}(\mathbf{x}) \equiv \max(1 - \mathbf{x}, \mathbf{x})$, as the top-left corner and the bottom-right corner of a cell also matches. Thus, the PatchMatch algorithm will choose the pixel based on both pixel similarity and regularity similarity to fill in the missing pixels (Fig. 4e).

## 4 Experiments

For evaluation, we introduce two datasets, and then apply box programs to vision and graphics tasks on these datasets, including plane segmentation (Sec. 4.1), image inpainting and extrapolation (Sec. 4.2), and view synthesis (Sec. 4.3).

**Dataset.** We collect two datasets from web image search engines for our experiments, a 44-image *Corridor Boxes* dataset and a 42-image *Building Boxes* dataset. These correspond to the inner view and the outer view of boxes, respectively. For both datasets, we manually annotate the plane segmentations by specifying edges of the boxes. For corridor images, we also create a mask for the far plane. For building images, we supplement the subject segmentation (i.e., the building of interest) to the dataset annotation.

### 4.1 Plane Segmentation

Because BPI develops a 3D understanding of the scene in the process of inferring the box program under perspective effects, an immediate application is single-image plane segmentation. Note that BPI performs this task by jointly inferring the perspective effects and the pattern regularity, in contrast to supervised learning approaches that train models with direct supervision, such as [22]. The core intuition is that the correct plane segmentation leads to the program that best describes each segmented plane in terms of repeated visual elements.

**Baselines.** We compare our method with two baselines: Huang et al. [16] and PlaneRCNN [22]. Huang et al. [16] first uses VLFeat [34] to detect vanishing points and line segments. It then generates a soft partition of the image into multiple parts by the support lines. PlaneRCNN is a learning-based method for plane segmentation. It uses a Mask-RCNN-like pipeline [14] and is trained on ScanNet [6].

**Metrics.** The output of each model is a set of masks indicating the segmented planes. We compare these with the ground-truth masks by computing the best match between two sets of masks, where the matching metric is the intersection over union (IoU). We then report the average IoU of all predicted masks. For corridor images, we exclude pixels in the far plane region during evaluation. Because we wish to use the plane segmentation to aid in image manipulation tasks such as image inpainting, we evaluate all methods on both original images (CrdO, BldO) and corrupted images (CrdC, BldC).

| Method | CrdO | CrdC | BldO | BldC |
|---|---|---|---|---|
| Huang et al. [16] | 0.30 | 0.30 | 0.73 | 0.69 |
| PlaneRCNN | 0.52 | 0.52 | 0.63 | 0.63 |
| BPI (Ours) | **0.84** | **0.84** | **0.86** | **0.86** |

Table 2: Plane segmentation results in IoU between detected and groundtruth planes. BPI outperforms both baselines on both original (CrdO, BldO) and corrupted images (CrdC, BldC).

**Results.** Fig. 5 and Table 2 show that BPI consistently outperforms the baselines on both corridor and building images. The baselines fail to detect planes when they contain complex structures and patterns.

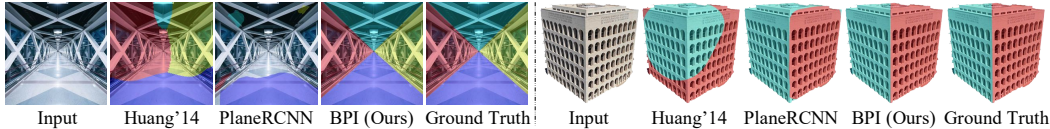

| | Input | Huang'14 | PlaneRCNN | BPI (Ours) | Ground Truth | | Input | Huang'14 | PlaneRCNN | BPI (Ours) | Ground Truth |

Figure 5: Visualization of the plane segmentation by different methods.

| | Corridors | | | | Buildings | | | |
|---|---|---|---|---|---|---|---|---|
| Method | $\mathcal{L}_1$ Mean ↓ | PSNR ↑ | SSIM ↑ | LPIPS ↓ | $\mathcal{L}_1$ Mean ↓ | PSNR ↑ | SSIM ↑ | LPIPS ↓ |
| PatchMatch | 53.12 | 31.55 | 0.9872 | 0.0215 | 34.88 | 32.99 | 0.9896 | 0.0072 |
| Image Melding | 58.50 | 30.54 | 0.9880 | 0.0174 | 49.10 | 30.42 | 0.9890 | 0.0134 |
| Huang et al. [16] | 51.88 | 31.41 | 0.9869 | 0.0177 | **26.10** | **34.87** | **0.9917** | **0.0049** |
| GatedConv | 44.98 | 32.32 | **0.9883** | **0.0153** | 55.31 | 29.54 | 0.9866 | 0.0112 |
| BPI (Ours) | **38.45** | **34.21** | **0.9892** | **0.0170** | 26.43 | 34.86 | 0.9913 | 0.0054 |

Table 3: We compare BPI-guided PatchMatch with both patch-based and learning-based methods on the task of image inpainting. ↑ indicates that the higher the number, the better. **Bold** indicates models that are indistinguishable with the best one under that metric with a linear mixed model. See text for details.

## 4.2 Image Inpainting

The inferred box programs support 3D-aware and regularity-preserving image manipulation, because they provide information on both what the perspective effects are and how the visual element repeats. To test BPI's performance on such tasks, we generate a dataset of corrupted images by randomly masking out regions of images from our two datasets.

**Metrics.** We use four metrics: pixel-level $\mathcal{L}_1$ distance, peak signal-to-noise ratio (PSNR), structural similarity index (SSIM) [36], and a learned perceptual metric LPIPS [44]. Following standard practice [27, 2], we compute PSNR and SSIM on image luma, and compute $\mathcal{L}_1$ distance and LPIPS directly on RGB values.

**Baselines.** We compare our model against both learning-based GatedConv [43] and three non-learning-based algorithms: PatchMatch [4], Image Melding [7], and Huang et al. [16]. All three non-learning algorithms are patch-based. Image Melding allows additional geometric and photometric transformations on patches. Huang et al. [16] first segments the image into different planes and augments a standard PatchMatch procedure with the plane rectification results.

**Results.** The results are summarized in Table 3 and Fig. 6. Here we run linear mixed models between each pair of methods and, for each metric, we mark in bold all methods that are indistinguishable with the best one. All $p$-values are in the supplementary material. Our method outperforms baselines on corridor images and achieves comparable results on building images with Huang et al. [16]. As discussed in Sec. 4.1, Huang et al. [16] relies on straight lines on the plane to segment and rectify the image, so it works well on planes with a plethora of such features. Huang et al. [16] tends to fail on images without dense straight lines on the plane (rows 3-4 of Fig. 6). On corridor images, beyond producing high-fidelity inpainting results, our regularity-aware PatchMatch process preserves the structure of the scene, such as the light on the ceiling in row 1 of Fig. 6.

**Image extrapolation.** Beyond inpainting missing pixels, our regularity-aware algorithm can extrapolate the box structure. Here, on the *Building Boxes* dataset, we show that our model can make the building taller or wider. The input to the model is a foreground mask of the building and the target region to be filled with the extrapolated building. We compare our method with four baselines: Content-Aware Scaling in Adobe Photoshop, Kaspar et al. [18], Huang et al. [16] and InGAN [32]. For content-aware scaling, we first select the foreground mask and then scale it so that it fills the target region. For both Kaspar et al. [18] and InGAN, we extract the bounding box of the foreground building and use it as the input. The model generates a new image that is 1.5x larger. For Huang et al. [16], we cast the extrapolation problem as inpainting the target region.

As shown in Fig. 7, Content-Aware Scaling is unaware of perspective effects and fails to preserve the lattice structure in the image. It also cannot generate new visual elements such as windows. Both Kaspar et al. [18] and InGAN do not preserve existing pixels when extrapolating the image. Kaspar et al. [18], as a texture synthesis method, also ignores the two-plane structure when generating the new

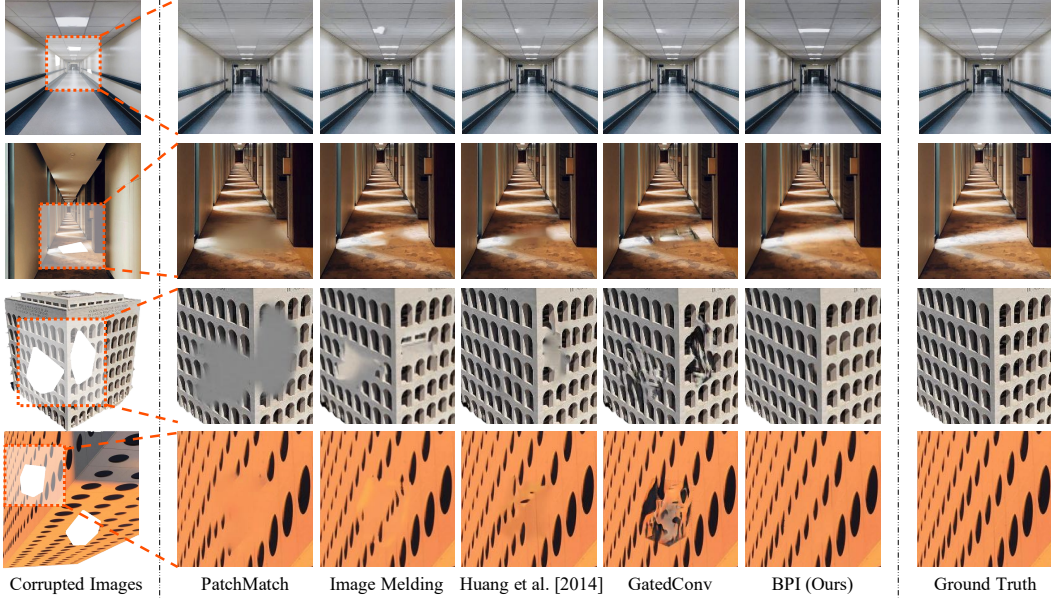

Corridor Images    PatchMatch    Image Melding    Huang et al. [2014]    GatedConv    BPI (Ours)    Ground Truth

Figure 6: Qualitative results on the task of image inpainting. Compared with the baselines, our model can better preserve the regular structures even if they are sparse or have strong perspective effects.

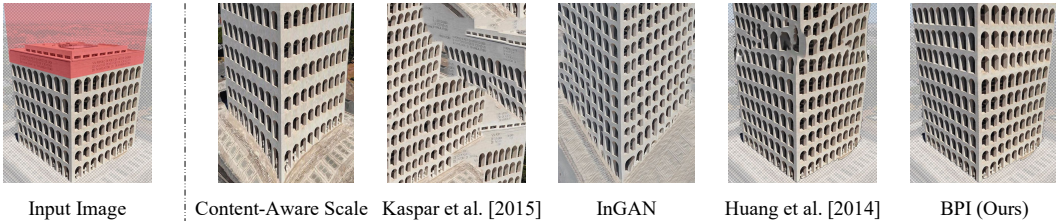

Input Image    Content-Aware Scale    Kaspar et al. [2015]    InGAN    Huang et al. [2014]    BPI (Ours)

Figure 7: The "extrapolated" buildings. Content-Aware Scale, Kaspar et al. [18], and InGAN fail to preserve the building structure while extrapolating the image: they either generate irregular patterns or change the shape of the planes. Huang et al. [16] fails to preserve the regular structure when inpainting large areas.

image. While InGAN is able to capture the visual elements, it does not follow the lattice pattern during synthesis. Huang et al. [16] can inpaint small local areas as in Fig. 6, but does not follow the regular structure when inpainting a large area. In contrast, BPI preserves both the plane structure of the building and the lattice pattern on each side.

Quantitatively, we randomly select 12 images, and ask 15 people to rank the outputs of different methods. We collected $12 \times 15 = 180$ responses. The preference for the models are: Ours(61%), Content-Aware Scaling(16%), Kaspar et al. [18] (2%), InGAN(16%), and Huang et al. [16] (5%).

### 4.3 View Synthesis

As our inferred box programs captures a holistic 3D scene representation, we can synthesize novel views of the scene from a *single* image. We compare different models on the *Corridor Boxes* dataset. We consider three types of camera movement: 1) "step into the corridor", 2) "step back in the corridor", and 3) "step back in the corridor, pan leftward, and tilt upward". All trajectories generate 5 frames. Detailed parameters are included in the supplemental material. Note that in both trajectories that involve "stepping back", the algorithm must synthesize pixels unseen in the original image. For our BPI, we run our program-guided PatchMatch to extrapolate the planes and synthesize pixels that are outside the input view frustum.

**Results.** We compare images generated by our BPI and by SynSin [37] in Fig. 8. The arrow on the input shows camera movement. As our method can generate a corridor of an arbitrary length, we see significantly fewer artifacts when the camera movement is large, compared with SynSin. Note that even in the first columns where the camera movement is small, the pixels synthesized by SynSin that are outside the original view frustum already fail to preserve the regular structure.

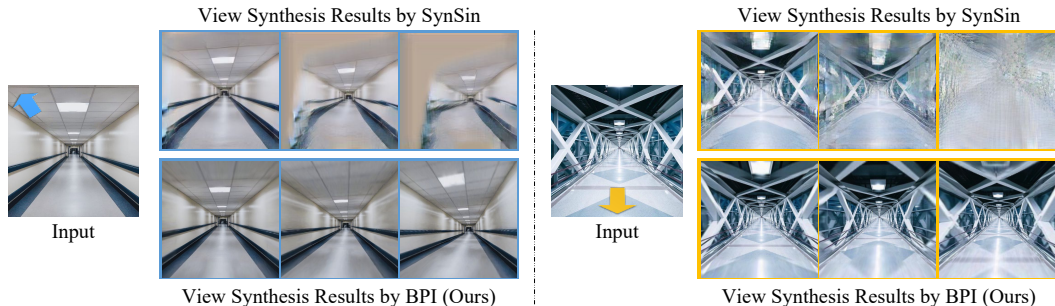

Figure 8: View synthesis from a single image of a corridor. Compared with the learning-based method SynSin, our model better preserves the regular patterns on the walls and also has remarkably fewer artifacts.

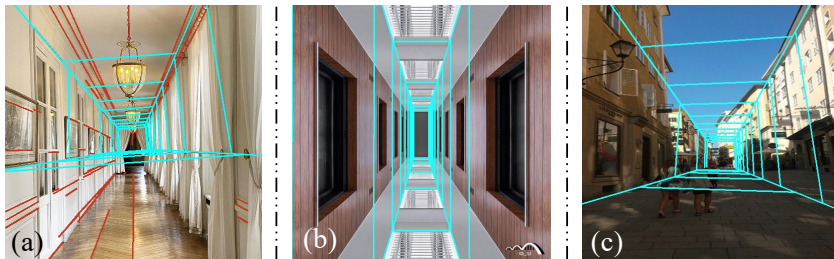

Figure 9: Failure cases. Our model may fail when neural networks misdetect visual cues (a). When image contains a solid color plane, our model may segment the pure white part to other planes (b). The inference might fail on irregular scenes, but could be mitigated with user interaction (c).

For a quantitative comparison, we randomly select 20 images, generate synchronized videos of the results produced by our method and by SynSin on all three camera trajectories, and ask 10 people to rank the outputs. We collected $20 \times 3 \times 10 = 600$ responses. For the three different trajectories, $100\%$, $94\%$, and $99.5\%$ of the responses prefer our result to that by SynSin, respectively.

### 4.4 Failure Case

Fig. 9 shows three main failure cases of our model. First, our model might misdetect vanishing points and wireframe segments, as illustrated in (a), where the model missed the wireframe between the floor and the right plane (detected wireframes shown as red lines). A second type of failure can occur when the image has a solid color plane. Illustrated in (b), our model segments the pure white part of the floor as part of the left/right planes. Finally, the inference might fail due to irregular planes, as shown in (c), where the buildings on the left do not form a rectangular plane. These issues could be mitigated with light user interaction, such as specifying wireframes.

## 5 Conclusion

We have presented Box Program Induction (BPI), a framework for inferring program-like representations that model the regular texture and patterns in 2D planes and the 3D posing of these planes, all from a single image. Our model assumes a *box prior*, which constrains the plane segmentation of the image, and uses visual cues to guide the inference. The inferred box program enables 3D-aware interactive image editing, such as inpainting, extrapolation, and view synthesis. Currently, our algorithm assumes that the full image can be partitioned into planes with regular structures. Future research may consider integrating models that can handle the presence of irregular image regions.

## Broader Impact

This paper presents an improved interactive image manipulation algorithms, which helps visual artists, photographers, and normal users who wants to perform content-aware and 3D-aware edits to their images. Moreover, our algorithm only use pixels from the input image itself during editing, which minimizes the biases coming from external sources. However, misuses of our algorithms can generate fake images that affect image forensics.

## Acknowledgements

This work is supported by the Center for Brains, Minds and Machines (NSF STC award CCF-1231216), NSF #1447476, ONR MURI N00014-16-1-2007, and IBM Research. Work was done while Jiajun Wu was a visiting researcher at Google Research.

## Footnotes

*: indicates equal contribution.

*Following common practice, we also fix other camera intrinsic properties: optical center to $(0, 0)$, skew factor to 0, and pixel aspect ratio to 1.

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
