[Supplementary Material · supp.pdf]

# Supplementary Material for
# Multi-Plane Program Induction with 3D Box Priors

We strongly recommend the reader to view the supplementary video, as it includes a graphical illustration of the Box Program Induction process and view synthesis animations.

This supplementary document is organized as the following. First, in Appendix A, we show the mathematical details of how to reconstruct the 3D positions and surface normal vectors of different planes based on the plane segmentation for a box's inner views (see Appendix B for outer views). Second, in Appendix B, we present the Box Program Induction (BPI) applied to *outer views* of boxes (details on BPI applied to *inner views* are in the main text). Next, in Appendix C, we discuss the implementation details of BPI. Finally, we present more qualitative results on box program induction, plane segmentation, image inpainting, and image extrapolation in Appendix D.

## A   Plane Reconstruction from Segmentation in *Inner Views*

In an inner view of a box, we use the plane segmentation of the input image to determine the 3D positions and surface normal vectors of four planes. The plane segmentation is represented as four rays starting from the detected vanishing point. Our reconstruction assumes a pinhole camera model with no lens distortion, as illustrated in Fig. 1a.

We start by defining the 3D coordinate system. We define the position of the camera pinhole $O$ as origin of the coordinate system. We also define the optical axis of the camera (i.e., the ray from the center of the image plane $\Pi'$ to $O$) as the $+z$ axis.

$V'$ denotes the vanishing point shown on the image plane. Four rays starting from $V'$ will intersect with the image boundary at four points $\{I'_k \mid k = 1, 2, 3, 4\}$. The line segments $\{V'I'_k \mid k = 1, 2, 3, 4\}$, namely the intersection line segments, are 2D projections of the intersection lines in 3D, between four planes. We also denote these 3D intersections lines as $\{I_k E_k \mid i = 1, 2, 3, 4\}$, where $I_k$ is the corresponding 3D projection of $I'_k$, and $E_k$ is the 3D projection of an arbitrary point on the 2D line segment $V'I'_k$. Thus, all lines $\{I'_k I_k \mid k = 1, 2, 3, 4\}$ should intersect at the optical center $O$.

As can be seen in Fig. 1a, the focal length (i.e. the distance between the image center and the optical center $O$) is correlated with the distance between $I_k$ and $O$ (i.e., the camera-to-subject distance)[*]. Moreover, the aspect ratio of the image sensor is correlated with the ratio between the sizes of four planes (i.e., the "aspect ratio" of the box). Thus, it is impossible to fully determine the focal length and the camera aspect ratio from this single image. Given this ambiguity, we assume the focal length to be 35mm and the aspect ratio to be 1, and then optimize for the equivalent distance between $I_k$ and $O$.

To this end, we first consider the following property of vanishing point: the line $V'O$ should be parallel with all 3D intersection lines $I_k E_k$. Thus, we perform an *orthographical projection* of the inner view using a new optical axis $V'O$. This leads to a new image $\Pi''$, as illustrated in Fig. 1b. The line segment $OI_k$ will also be projected onto $\Pi''$ as a line segment $OI''_k$, and four intersection lines $I_k E_k$ will become four points on $\Pi''$. Determining the distance between $O$ and $I_k$ is equivalent to determining the distance between $I''_k$ and $V'$ on the new image plane $\Pi''$.

---

[*]Formally, we need three planes that are perpendicular to each other to determine the focal length based on the perspective effect. However, we have only two such planes here. See Liebowitz et al. [5] for a detailed discussion.

(a) Perspective projection in an *inner view*       (b) Orthographic projection from the vanishing point

Figure 1: Illustration of (a) the perspective projection in an inner view of the box (image upside down to be consistent with the projection), and (b) the orthographic projection centered at the vanishing point.

Unfortunately, one can not fully determine the distance between $I_k''$ and $V'$, but only the ratio between $V'I_1''$ and $V'I_k''$, $k = 2, 3, 4$, even if we have assumed the focal length and the aspect ratio. This is because this camera-to-subject distance is also correlated with the actual size of the box. Intuitively, a box that is close and small may look identical in the image as another box that is far but big. Thus, we will manually set the distance between $V'$ and $I_1''$ to be 1 meter. It is important to note that, although we have manually set the focal length, the aspect ratio, and the camera-to-subject distance, these value of these parameters will not affect the plane rectification results. Moreover, they also have no influence on downstream tasks such as image inpainting.

With the position of $I_1''$, we then determine the positions of $I_k''$, $k = 2, 3, 4$, relative to the position of $I_1''$. We use the following *box prior*: four planes of the box are either perpendicular or parallel to each other. Thus, on the orthographic image plane $\Pi''$, the quadrilateral $I_1''I_2''I_3''I_4''$ should be a rectangle.

Now we use the 2D coordinates defined on the plane $\Pi''$ and centered at the vanishing point $V'$. Denote the coordinates of $I_k''$ on $\Pi''$ by $(x_k, y_k)$, the slope of $I_1''I_2''$ by $u$, and the slope of $I_2''I_3''$ by $v$. Also denote the slope of $VI_k''$ by $w_k$, $k = 1, 2, 3, 4$, as illustrated in Fig. 1b. We have the following equation system:

$$\begin{cases} y_k = w_k x_k; \ k = 1, 2, 3, 4 & (I_k'' \text{ lies on the ray } VI_k''.) \\ y_1 - y_2 = u(x_1 - x_2) & (\text{definition.}) \\ y_2 - y_3 = v(x_2 - x_3) & (\text{definition.}) \\ y_3 - y_4 = u(x_3 - x_4) & (I_1''I_2'' \text{ is parallel to } I_3''I_4''.) \\ y_4 - y_1 = v(x_4 - x_1) & (I_2''I_3'' \text{ is parallel to } I_4''I_1''.) \\ x_1^2 + y_1^2 = 1 & (\text{camera-to-subject distance assumption.}) \\ uv = -1 & (I_1''I_2'' \text{ and } I_2''I_3'' \text{ are perpendicular to each other.}) \end{cases}$$

This equation system allows us to solve for $u$ and $v$ unambiguously. In fact, there exists a closed-form solution to the values of $u$ and $v$, which is independent of $(x_k, y_k)$. After determining $u$ and $v$, we can further compute the positions of $I_k''$ and thus the 3D positions of $I_k$, $k = 1, 2, 3, 4$.

During inference, we first compute the orthographic projection $\Pi''$ based on the detected vanishing point and the focal length. Next, by fixing the position of $I_1''$, we solve for the other $I_k''$, $k = 2, 3, 4$ on the orthographic image. Finally, we project the solution back to the 3D space and determine the 3D positions and surface normal vectors for individual planes.

## B   Box Program Induction for *Outer Views* of Boxes

In this section, we present the box program induction for *outer views* of boxes. The whole process is almost identical to the *inner view* case, except that for an outer view, we only need to consider

two planes (e.g., two side planes of a building), with the other planes being either non-visible or foreshortened severely. The full search process consists of four steps. First, we use pre-trained neural networks to detect the 2D wireframe line segments (but no vanishing points) from the image. We also filter out wireframe segments that are too short in length. Next, we generate a set of candidate plane segmentation maps by partitioning the image based on the detected wireframe segments. Then, for each segmented plane, we seek the program that best describes the regularity on each plane. Finally, we rank all candidate plane segmentations by the fitness score (defined in the main text).

**Step 1: Visual cue detection.** Following the *box prior*, an outer view of a box contains only two planes. Therefore, there will be only one intersection line between these two planes (e.g., two walls of a building). We use L-CNN [8] to extract wireframes in the image. Next, we filter out wireframe segments whose length is smaller than a threshold $\delta_1$. The remaining wireframe segments are denoted by the set $WF$. Note that unlike the inner view case, we do not use vanishing point detection for outer views.

**Step 2: Plane segmentation.** Next, we then extend every wireframe segment to a line. Each line will partition the input image into two parts, which we treat as the candidate plane segmentation of the image.

**Step 3: Plane rectification and regularity inference.** Since we only have two planes, we cannot use the plane segmentation to fully reconstruct the positions and surface normal vectors of different planes. We run the Perspective Plane Program Induction (P3I) algorithm [4] on each plane to jointly infer the surface normal of each plane and its regularity structure.

**Step 4: Box program ranking.** Identical to the inner view case, we sum up the fitness score for two planes in each candidate segmentation as the overall program fitness. We use this score to rank all candidate segmentations, and pick the program with the highest fitness to describe the entire image.

## C   Implementation Details

When filtering wireframes by length, we set $\delta_1 = 0.1 \times \min(w, h)$, where $w, h$ are the width and height of the input image, respectively. Radius used to filter wireframes toward the detected vanishing point is $\delta_2 = 0.01 \times \min(w, h)$. In the plane rectification step, we rectify the plane region to a $200 \times 200$ image, on which we infer the regularity. We assume that objects repeat at least 3 times on each plane.

## D   Additional Results

**Time complexity.** For the corridor dataset, statistically, each image contains 1,506 wireframe combinations on average. 46 programs are evaluated on each plane. Note that BPI reduces the search space significantly based on the box prior, so that the search can be done efficiently ($23\times$ faster than without the box prior).

We also show the runtime of different algorithms on the task of plane segmentation (Table 1) and image inpainting (Table 2). The Image Melding [2] and Huang et al. [3] baselines are tested on a single machine with an Intel i7-6500U@2.5GHz CPU and 8GB RAM. All other baselines are tested on a single machine with an Intel E5-2650@2.20GHz CPU, a GeForce GTX 1080 GPU, and 8GB RAM.

**Qualitative results.** We supplement more results on box program induction (Fig. 2). Our model can be applied to less constrained images by allowing the user to specify the regular region. It also works for images where not all planes of a box exhibit regular patterns. Fig. 2 (i) shows example results. In (a), we run BPI on a user-specified region (the orange bounding box). BPI outputs a reasonable program even though the left plane is curved. We also show that our model can be applied to a broader class of images than buildings, such as the bamboo forest in (b), and the scene with irregular planes in (c).

We also provide more qualitative results for plane segmentation (Fig. 3), image inpainting (Fig. 4) and image extrapolation (Fig. 5).

| | Huang et al. [3] | PlaneRCNN [6] | BPI (Ours) |
|---|---|---|---|
| Corridor Boxes | 1.91s | 0.14s | 43.50s |
| Building Boxes | 6.83s | 0.76s | 171.20s |

Table 1: Runtime of different methods on the task of plane segmentation.

| | PatchMatch [1] | Image Melding [2] | Huang et al. [3] | Gated Conv [7] | BPI (Ours) |
|---|---|---|---|---|---|
| Corridor Boxes | 61.6s | 1275.1s | 18.9s | 1.9s | 20.1s |
| Building Boxes | 122.1s | 808.3s | 64.4s | 4.9s | 41.4s |

Table 2: Runtime of different methods on the task of image inpainting.

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

```
SetPlane_l(pos=[-1.4, -1.48, 1.68],
    normal=[0.40, 0.06, 0.92])
for i in range(0, 6)
  for j in range(0, 9)

    Draw(        , x = -28 * i
                    +   6 * j,
                 y =  52 * j)
SetPlane_r(pos=[-1.5, -1.49, 1.10],
    normal=[-0.07, 0.14, 0.99])
for i in range(0, 5)
  for j in range(0, 5)

    Draw(        , x = 55 * i,
                 y = 52 * j)
```

```
SetPlane_l(pos=[-1.1, -1.39, 0.23],
    normal=[-1.00, -0.01, -0.00])
for i in range(0, 10)
  Draw(        , x = 20 * i)
SetPlane_t(pos=[1.04, -1.37, 0.23],
    normal=[0.01, -0.95, -0.31])
for i in range(0, 10)
  Draw(        , x = 20 * i)
SetPlane_r(pos=[1.00, 1.55, 1.19],
    normal=[1.00, 0.01, 0.00])
for i in range(0, 10)
  Draw(        , x = 20 * i)
SetPlane_b(pos=[-1.15, 1.52, 1.19],
    normal=[-0.01, 0.95, 0.31])
for i in range(0, 10)
  Draw(        , x = 20 * i)
```

```
SetPlane_l(pos=[-1.0, -2.77, 0.10],
    normal=[-1.00, -0.01, 0.03])
for i in range(0, 10)
  Draw(        , x = 20 * i)
SetPlane_t(pos=[1.03, -2.75, 0.03],
    normal=[0.01, -0.99, -0.10])
for i in range(0, 3)
  Draw(        , x = 65 * i)
SetPlane_r(pos=[1.01, 0.42, 0.36],
    normal=[1.00, 0.01, -0.03])
for i in range(0, 9)
  Draw(        , x = 22 * i)
SetPlane_b(pos=[-1.02, 0.40, 0.42],
    normal=[-0.01, 0.99, 0.10])
for i in range(0, 6)
  Draw(        , x = 30 * i)
```

Figure 2: More image examples and the corresponding programs. We use cyan lines to visualize the lattice structure on each plane.

| Input | Huang'14 | PlaneRCNN | BPI (Ours) | Ground Truth |

Figure 3: Visualization of the plane segmentation by different methods.

Figure 4: Qualitative results on the task of image inpainting.

| Corrupted Images | PatchMatch | Image Melding | Huang et al. [2014] | GatedConv | BPI (Ours) | Ground Truth |

| Input Image | Content-Aware Scale | Kaspar et al. [2015] | InGAN | Huang et al. [2014] | BPI (Ours) |

Figure 5: Qualitative results on the task of image extrapolation.