[Reviews · NeurIPS 2020]

Review 1

Summary and Contributions: The paper proposed BPI, Box Program Induction, a framework that can jointly infer the planes with it's positions and orientations, repeated structures, and camera poses from single images. Experiments on image inpainting and extrapolation, and view synthesis, demonstrated the usefulness of proposed approach.

Strengths: The paper provides a full pipeline to infer the 3D scene structure from single images with box priors, by assuming the images are formulated from the interior of a 3D box, or the exterior. In experiments, the generated novel views are realistic and better than baselines.

Weaknesses: 1. The applicability of the proposed BPI is restricted. Although I agree infering 3D scene geometry is an important task for 3D image understanding, I'm wondering how does the proposed method generalize to real scene, where the images are not essentially generated from either inside of a "3d box", nor outside of a "3d box" (e.g. Fig. 9 in NeurVPS[45]). More importantly, the fitting score (which is used for ranking different programs) is defined based on the repeated patterns in the plane image, but even in many images that are formed from a 3d box does not necessarily have repeated patterns (such as indoor bedroom), then how does the proposed method work on this case (where the proposed fitting score may not be meaningful)? 2. The author assumed the pattern should be appearing in lattice, can it handle arbitrary placed repeated patterns (only assuming they did not overlap)? 3. Searching for the program would not only require iterate different wireframe combination, but also, for the same wireframe combination, it requires to iterate all different number of columns and number of rows for the repeated patterns in each plane, could the author provide how big this could be and how fast it would run?

Correctness: Yes, the claims and method are correct.

Clarity: Yes, the paper is overall well written. Though I think there are some places unclear (please see weakness session).

Relation to Prior Work: Yes, the paper made a good discussion with prior works.

Reproducibility: Yes

Additional Feedback: Overall, although I agree the target task is meaningful and thinkingful for 3d image understanding, the proposed approach has many restrictions (Please see weakness section), so I vote for a weak reject initially, I'd like to listen to author's rebuttal. #######POST REBUTTAL COMMENTS Thanks for the rebuttal, it addressed most of my concerns, and I am willing to upgrade my score to 6.


Review 2

Summary and Contributions: This paper extends the inverse graphics for scene understanding direction from Learning to Describe Scenes with Programs, ICLR 2019 and P3I (Perspective Plane Program Induction), CVPR 2020. They add an additional layer of complexity of detecting multiple planes in an image (with a box prior, e.g. there are 4 major planes when you view a box from the inside) and then using a method similar to P3I to “explain” the regularity in every plane with a program. Finding a best fit program helps filter candidate perception outputs improving its performance (here, plane detection) and allows program guided image manipulation, such as inpainting, extrapolation and view synthesis (similar to the idea presented in Program Guided Image Manipulators, ICCV 2019 and uses in P3I), but done using a PatchMatch formulation that additionally uses the predicted regularity pattern in the image. Results on two self collected datasets show significant improvements in these use-cases over baselines as well as over recently proposed methods.

Strengths: - Writing: The paper is very clearly written and is easy to follow and understand - Concrete Progress: This direction of improving perception and allowing manipulation through search based program induction is in my opinion extremely important for making strong neuro-symbolic models that explain the visual world. This paper makes progress in this direction over P3I by adding plane detection and explaining individual planes that enables this kind of scene understanding on a larger subset of natural images than was possible in P3I - Strong Results: On the kinds of images that this method can model (buildings / corridors with strong regularity cues), it beats existing state-of-the-art baselines on image extrapolation, inpainting and view-synthesis by a very convincing margin, both qualitatively, quantitatively and with human studies.

Weaknesses: - More results showing inferred programs: The paper lacks qualitative results of inferred programs on images. To the reader, it would be very interesting to understand what the inferred program for images look like for example. For example, being able to see the inferred program for the right plane in Figure 6. Row 1 would be interesting since it is a continuous repeating texture without distinct repeating points. What is the range in the for loop that is found here? - Where does this fail?: What are the results for inpainting, extrapolation, view synthesis where humans did not prefer your paper’s results? What are the failure modes? These have been completely missed in the paper and is very important to add. I believe this shouldn’t be too much of a hassle for the authors to add, because of which I am not changing my rating. Without these results being presented in the revision however, I would definitely drop my final rating. Similarly, how does the method perform beyond the images with strong regularity cues that were collected (for example a building/corridor without regularity)? Does it fail gracefully? What do the inferred programs look like? - “Inferring camera”: If I understand correctly, fixed intrinsics and extrinsics are assumed for the camera and all plane parameters are predicted in that fixed coordinate frame. The authors promise 3D inference for the camera as part of the program induction, which I believe is slightly different from what is proposed. Since it’s not like the 3D representation is predicted in a canonical view and the camera is inferred w.r.t. some canonical view, the writing is misleading in my point of view. I would be more comfortable with “our inference works with fixed camera intrinsics / extrinsics in the scene” instead of “we infer the camera parameters"

Correctness: Yes, I believe so. I would re-visit the terminology for “inferring the camera”, but at the end of the day it is irrelevant to the task being done.

Clarity: Yes, the paper is written clearly. It does assume knowledge of some previous papers, but I cannot see a way of putting all of this information into 8 pages while also going into details on them.

Relation to Prior Work: Yes, the paper is positioned very clearly with respect to prior work.

Reproducibility: Yes

Additional Feedback: - I would improve the writing w.r.t. camera inference in the paper. It doesn’t take away any credit from what the proposed method can achieve, but promises something different in the introduction in my opinion. I am also very open to hearing the authors’ rebuttal on this point of view - I would add lots of negative results to go with the great results you demonstrate in the paper, along with adding more examples of inferred programs on images. I would also add results on images outside the subset of images which can be modelled by the method. I am looking forward to seeing some of these in the rebuttal. -------------- While this work is not a wildly new idea given the previous papers mentioned in the review, it is indeed a non-trivial extension that extends the frontier of the analysis by synthesis by exploiting regularity for visual scene understanding. It would be great to see how the method works on images which do not fall into the category of images this method can model. This understanding is important for researchers to be able to see its failure modes and decide on future research directions.


Review 3

Summary and Contributions: This paper continues a line of work (e.g. [11, 23, 24]) where images are modeled in terms of programs, in which a set of elementary components and a grammar for those are retrieved from the image. This can be used for image manipulation such as inpainting. Computational complexity has been a strong limiting factor, confinining earlier methods to operate on highly specific domains or drawings. The contribution here is that the 3D structure of the scene is modeled in addition to the appearance. Whereas previous methods operated in the image plane, the current method assumes that the scene is constituted by a box with perpendicular flat surfaces, either viewed from the inside or the outside. Another assumption is that there is some regularity to the pattern on each surface. This strong prior allows scaling up the program reasoning, making it possible to address much more complex scenes. Experiments show that the proposed method greatly outperforms the state-of-the-art both in plane segmentation and inpainting for scenes with box structure and regular patterns on the surfaces.

Strengths: This is an excellent paper, with a clear line of argument, theoretically grounded method and clear motivation of the design choices in relation to the state of the art. The contribution is significant in that the method is simple and robust, yet highly accurate, and that it operates on only the present image - which is an advantage for security reasons as outlined in the Broader Impact section. The experiments are well designed and valid.

Weaknesses: The obvious limitation is that the method is only applicable for scenes which are dominated by a box structure with regular surface patterns, i.e., man-made structures, primarily the in- and outside of buildings. However, this is clearly communicated in the paper.

Correctness: Both the claims and the method, as well as the methodology of evaluation, are correct as far as I can judge.

Clarity: The paper is very well written, with good organization, a clear line of reasoning and good writing and language structure (including paragraph structure which allows speed-reading and descriptive section and paragraph headings).

Relation to Prior Work: The paper clearly describes the relation to: - earlier visual program induction work, including top-down 3D reasoning (which is limited to simple scenes) and combinations of top-down and bottom-up reasoning/learning (which has been focusing on models with a single image plane), - earlier image manipulation work, including earlier methods (which fail to model high-level and structural info) and deep-learning based methods (which require a volume of training images). The relations to [11, 20, 23, 24] is also discussed in more detail.

Reproducibility: Yes

Additional Feedback: WRITTEN AFTER REBUTTAL: After taking part of the other reviews and the rebuttal, I would like to maintain my overall score.


Review 4

Summary and Contributions: This paper presents a program induction method in 3D space, where a scene representation is inferred to model repeated structure on 2D planes, in addition, 3D position and orientation and camera parameters can be inferred all from a singe image. Experiments of different tasks such as plane segmentation, image inpainting, extrapolation view synthesis are conducted on two datasets, including Corridor Boxes and Building Boxes collected by the authors.

Strengths: This method can infer a lot of information such program-lie scene representation, 3D position, orientation and camera parameter all from one image. This method have potentials to be implemented to many tasks.

Weaknesses: -I cannot follow the technical novelty and contribution of this paper. -It seems PlaneRCNN [21] is designed for more general cases, however, the proposed method benefits from a 3D Box priors, when evaluating on 3D box like scenes, it seems unfair to [21]. Include 3D priors to [21] seems fairer. -Compared [21], it seems this work is limited to box structures, I’m not sure the practicability of this method, and how confidential can we obtain 3D priors in real world. -It seems [42] is also designed for more general and complex cases without 3D box priors, current evaluation and comparison seem not capable of demonstrating the effectiveness of the proposed method.

Correctness: It should be correct.

Clarity: The writing is acceptable.

Relation to Prior Work: Yes, this issue has been clearly discussed. Some problems with related work pls see the weakness part.

Reproducibility: Yes

Additional Feedback:


Review 5

Summary and Contributions: Update: Thanks to the authors for their feedback. I didn't see any concerns from the other reviewers that would cause me to change my rating. ======== This work introduces “Box Program Induction” which uses a structured model of 2D surfaces with repeated structures, their 3D orientation/position, and the camera model. They infer this from a single image. This is derived from a deep-network that infers cues like vanishing points and “wireframe lines” that guide a search to find a program that best explains the image. While there as been extensive work in inferring the structure of box-like scenes, the modeling of the appearance, especially repetitive structures, of these scenes appears novel. Note that this modeling process appears closely related to [20]. The main contribution is the extension of these plane-based models to box structures. A deep network predicts vanishing points and wireframe (a collection of lines and junctions) from [45], and [46] respectively. Then they propose candidate plane segmentations by using the wireframe and updated vanishing points and the box assumption. Under this box constraint, they can estimate the 3D positions and orientations of the planes. Finally, they rectify the planes and create a plane program for each. Each candidate box program is scored by the sum of the plane fitness scores. Finally, they introduce a “program-guided” extension to PatchMatch to enable the image manipulation operations. The plane program provides a regular lattice which can be used to emphasize patches that have similar appearance at similar relative positions in their respective lattice cells. To support their experiments, they introduce a dataset with 42 images of outer-views of boxes and 44 images of inner-views of boxes. They test on plane segmentation, inpainting, extrapolation (i.e. zooming out), and viewpoint synthesis. They explore a variety of baselines, outperforming them across the board. This should be expected because a number of the approaches weren’t designed for this specific application. Although the work relies heavily on prior work, the new capabilities are compelling. The application of these existing ideas to enable new capabilities should be interesting to this audience (although might be better suited for a computer vision conference).

Strengths: - As noted above the main strength of this work is that it takes advantage recent work on plane programs, and extends to 3D box structures to enable a variety of image manipulation applications. The results are compelling for the dataset provided.

Weaknesses: - One downside of this work is that it leans heavily on prior work. The heavy lifting is done by the planar programs of [20], and the layout cue inference from [45] and [46]. However, the combination leads to interesting new capabilities. - Since this method relies on some pretty heavy assumptions, it would be nice to see an exploration of what will make it fail.

Correctness: Appears to be ok.

Clarity: Overall the paper is well organized and written.

Relation to Prior Work: There is a wide range of literature on inferring room layout from a single image that relies on similar cues like vanishing lines and box priors. One popular example is Recovering the spatial layout of cluttered rooms by Hedau et al.

Reproducibility: Yes

Additional Feedback:

[Author Response · NeurIPS 2020]

We thank all reviewers for their comments. We begin with general responses, followed by specific ones.

**R1,R3,R4,R5: More results on general images.** Our model can also be applied to less constrained images by allowing
the user to specify the regular region. It also works for images where not all planes of a box exhibit regular patterns.
Fig. 1 (i) shows example results. In (a), we run BPI on a user-specified region (the orange bounding box). BPI outputs a
reasonable program even though the left plane is curved. We also show that our model can be applied to a broader class
of images than buildings, such as the bamboo forest in (b), and the scene with irregular planes in (c).

**R3,R6: Failure cases.** Failure modes can be roughly divided into three categories, shown in Fig. 1 (ii). First, our model
might misdetect vanishing points and wireframe segments, as illustrated in (d), where the model missed the wireframe
between the floor and the right plane (detected wireframes shown as red lines). A second type of failure can occur when
the image has a solid color plane. Illustrated in (e), our model segments the pure white part of the floor as part of the
left/right planes. Finally, the inference might fail due to irregular planes, as shown in (f), where the buildings on the
left do not form a rectangular plane. These issues could be mitigated with light user interaction, such as specifying
wireframes. We will add more failure modes to the revised paper.

**R3,R4,R5: Contribution and baselines.** Our main contribution is to jointly model 3D structure and program-like,
repeated patterns and use the inferred structure for image manipulation. We achieve high-fidelity and regularity-
preserving image editing by exploiting these structures and regularities. Our model outperforms existing methods for
general image editing (GatedConv) and also models relying on similar assumptions (Huang et al. [2014]).

**R1: Non-lattice patterns.** Non-lattice patterns can be handled the same way as in P3I [20]. Specifically, we can extend
our DSL to include other types of patterns (e.g., circular, and radial), and apply the same inference algorithm.

**R1: Search space and speed.** For the corridor dataset, each image contains 1,506 wireframe combinations on average.
46 programs are evaluated on each plane. The full process takes 196s. Note that BPI reduces the search space significantly
based on the box prior, so that the search can be done efficiently ($23\times$ faster than without the box prior).

**R1: The inferred program for Figure 6 Row 1.** The program for this image can be found in Figure 1 of the main
paper. Note that the side planes in this image are not continuous repeating textures due to the lights on the ceiling.

**R3: Camera parameters.** Thanks! We will revise our wording to "works with fixed camera intrinsics and extrinsics".
It is worth noting that while it's intrinsically ambiguous to infer camera parameters from a single picture, as they
are entangled with the pose of the box or the distance to the object, different sets of camera intrinsics turn out to be
equivalent when it comes to image synthesis and manipulation. Even if the assumed camera parameters are different
from the ground truth values of the image, it won't affect the produced pictures. We will supply a derivation in the supp.

**R4: PlaneRCNN with 3D prior.** We add an extra experiment for PlaneRCNN with the box prior. Specifically, we
post-process the output of PlaneRCNN and seek mutually perpendicular planes, using the surface normals. This
improves the segmentation IoU from 0.52 to 0.58, evaluated on the corridor dataset. Our result is 0.84.

**R5: Related work.** We will cite and discuss the referred paper. Unlike the suggested paper, we focus on the *joint*
inference of 3D structures and programmatic patterns, with applications to image synthesis.

(i) General Cases          (ii) Failure Cases

Figure 1: More image examples and the corresponding programs. We use cyan lines to visualize the lattice structure on each plane.

[Meta-Review · NeurIPS 2020]

This paper is about describing scenes as programs, and in fact inferring the program that generated a scene, given a formal grammar. It relies heavily on previous work [11,23, 24] in terms of program induction for scene generation, and it is concerned exclusively with box-shaped scenes (buildings, indoor spaces, etc). This can be used for image manipulation such as inpainting. As R4 points out, "the contribution here is that the 3D structure of the scene is modeled in addition to the appearance. Whereas previous methods operated in the image plane, the current method assumes that the scene is constituted by a box with perpendicular flat surfaces, either viewed from the inside or the outside. Another assumption is that there is some regularity to the pattern on each surface. This strong prior allows scaling up the program reasoning, making it possible to address much more complex scenes. Experiments show that the proposed method greatly outperforms the state-of-the-art both in plane segmentation and inpainting for scenes with box structure and regular patterns on the surfaces." I think this description is very accurate. This paper was well-received. The initial concerns about how the method would do if applied to non-regular scenes, like the ones assumed in the paper, was addressed by the authors in the rebuttal phase. I think despite the narrow scope of the paper (box scenes), reviewers liked the idea of doing program induction that involves both 3D structure and 2D observations, and they recommend acceptance. I agree with this assessment and I suggest a poster.